# A DNase Type VI Secretion System Effector Requires Its MIX Domain for Secretion

Chaya Mushka Fridman,[a] Biswanath Jana,[a] Rotem Ben-Yaakov,[a*] (ID) Eran Bosis,[b] (ID) Dor Salomon[a]

[a]Department of Clinical Microbiology and Immunology, Sackler Faculty of Medicine, Tel Aviv University, Tel Aviv, Israel
[b]Department of Biotechnology Engineering, Braude College of Engineering, Karmiel, Israel

Chaya Mushka Fridman and Biswanath Jana contributed equally to this article. Author order was determined on the basis of earlier involvement in the project.

**ABSTRACT** Gram-negative bacteria often employ the type VI secretion system (T6SS) to deliver diverse cocktails of antibacterial effectors into rival bacteria. In many cases, even when the identity of the delivered effectors is known, their toxic activity and mechanism of secretion are not. Here, we investigate VPA1263, a *Vibrio parahaemolyticus* T6SS effector that belongs to a widespread class of polymorphic effectors containing a MIX domain. We reveal a C-terminal DNase toxin domain belonging to the HNH nuclease superfamily, and we show that it mediates the antibacterial toxicity of this effector during bacterial competition. Furthermore, we demonstrate that the VPA1263 MIX domain is necessary for T6SS-mediated secretion and intoxication of recipient bacteria. These results are the first indication of a functional role for MIX domains in T6SS secretion.

**IMPORTANCE** Specialized protein delivery systems are used during bacterial competition to deploy cocktails of toxins that target conserved cellular components. Although numerous toxins have been revealed, the activity of many remains unknown. In this study, we investigated such a toxin from the pathogen *Vibrio parahaemolyticus*. Our findings indicate that the toxin employs a DNase domain to intoxicate competitors. We also show that a domain used as a marker for secreted toxins is required for secretion of the toxin via a type VI secretion system.

**KEYWORDS** DNase, T6SS, *Vibrio*, competition, effector, secretion systems

**B**acterial polymorphic toxins are modular proteins delivered by diverse secretion systems to mediate antibacterial or anti-eukaryotic activities (1, 2). They often share an N-terminal domain fused to diverse C-terminal toxin domains. The N-terminal domain, which can be used to classify these toxins, may determine which secretion system the effectors will be secreted through, e.g., the type V, VI, or VII secretion systems (T5SS, T6SS, or T7SS, respectively) (1, 3–6).

Many polymorphic toxins are secreted via T6SS, a contractile injection system widespread in Gram-negative bacteria (7, 8). The toxins, called effectors, decorate a secreted arrow-like structure comprising an inner tube, made of stacked Hcp hexamers, and a capping spike containing a VgrG trimer and a proline-alanine-alanine-arginine (PAAR) repeat-containing protein (9–12); the effector-decorated arrow is propelled outside the cell by a contracting sheath structure that engulfs the inner tube (13). Effectors can be either specialized—Hcp, VgrG, or PAAR proteins containing additional toxin domains at their C termini—or cargo effectors, which are proteins that noncovalently bind to Hcp, VgrG, or PAAR with or without the aid of an adaptor protein or a co-effector (14–21). To date, three classes of polymorphic T6SS cargo effectors have been characterized, containing MIX (3), FIX (22), or Rhs (23) domains; the Rhs domain is not restricted to T6SS effectors (23).

**Ad Hoc Peer Reviewer** (ID) Erh-Min Lai, Academia Sinica

Address correspondence to Dor Salomon, dorsalomon@mail.tau.ac.il.

*Present address: Rotem Ben-Yaakov, TransAlgae, Rehovot, Israel.

The authors declare no conflict of interest.

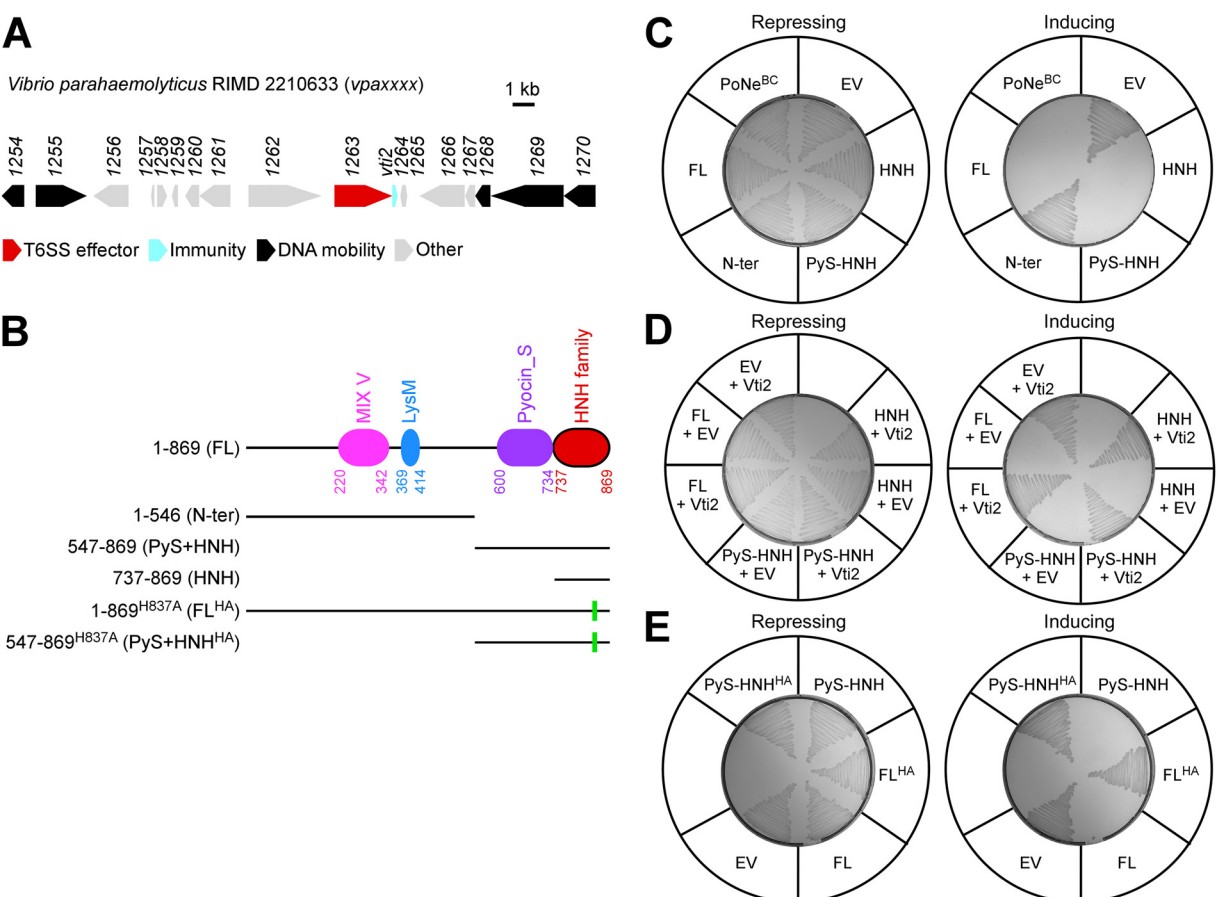

**FIG 1** C-terminal domain mediation of VPA1263's antibacterial toxicity. (A) Neighborhood of the *vpa1263* gene (GenBank accession number BA000032.2). Genes are denoted by arrows indicating the direction of transcription. Locus tags (*vpaxxxx*) and unannotated gene names (i.e., *vti2*) are shown above the arrows. (B) Schematic representation of the domains identified in VPA1263 and of truncated and mutated variants examined in this figure. Green rectangles denote the position of the H837A mutation. (C to E) Toxicity of C-terminal Myc-His-tagged VPA1263 variants expressed from an arabinose-inducible expression plasmid in *E. coli* MG1655 streaked onto repressing (glucose) or inducing (arabinose) agar plates. In panel D, a second arabinose-inducible plasmid was used to coexpress Vti2. The results shown represent at least three independent experiments. EV, empty plasmid; PoNe^BC, a PoNe domain-containing DNase from *Bacillus cereus* (BC3021); N-ter, N-terminal region; FL, full-length protein; FL^HA, FL with conserved histidine 837 replaced by alanine; PyS, Pyocin_S domain; HNH, HNH family domain.

Proteins belonging to the MIX effector class are secreted via T6SS (3, 24–26). They contain a predominantly N-terminal MIX domain fused to known or predicted anti-eukaryotic or antibacterial C-terminal toxin domains; the latter precede a gene encoding a cognate immunity protein that prevents self- or kin intoxication (4, 27, 28). MIX domains can be divided into five clans that differ in their sequence conservation (3, 29). Notably, members of the MIX V clan are common in bacteria of the *Vibrionaceae* family (3, 29) and were suggested to be horizontally shared via mobile genetic elements (25). It remains unknown whether MIX plays a role in T6SS-mediated secretion.

In previous a work, we identified the MIX effector VPA1263 as an antibacterial effector delivered by *V. parahaemolyticus* RIMD 2210633 T6SS1 and its downstream encoded Vti2 as the cognate immunity protein (3) (Fig. 1A). VPA1263 belongs to the MIX V clan. It is encoded on *V. parahaemolyticus* island 6 (VpaI-6), a predicted mobile genomic island that is present in a subset of *V. parahaemolyticus* genomes (30). In this work, we aimed to investigate the toxic activity and the secretion mechanism of VPA1263. We found that VPA1263 contains a C-terminal toxin domain belonging to the HNH nuclease superfamily and showed that this domain functions as a DNase. Furthermore, we found that the MIX domain is required for secretion of VPA1263 via the T6SS, providing the first experimental validation of the hypothesis that MIX domains play a role in T6SS effector secretion.

## RESULTS

**VPA1263 contains a C-terminal HNH nuclease-like toxin domain.** Analysis of the amino acid sequence of VPA1263 using the NCBI conserved domain database (31) revealed three known domains: a MIX domain (3), a peptidoglycan-binding LysM domain (32), and a Pyocin_S domain (33) (Fig. 1B). Hidden Markov modeling using HHpred (34) revealed another region at the C terminus of VPA1263 (amino acids 737 to 869) that is similar to the toxin domain found in members of the HNH nuclease superfamily, such as antibacterial S-type pyocins and colicins (35–40). An iterative search using a hidden Markov model profile against the UniProt protein database indicated that the Pyocin_S and putative HNH nuclease domains located at the C terminus of VPA1263 are also found together at the C termini of specialized T6SS effectors containing PAAR, Hcp, or VgrG (see Fig. S1A in the supplemental material); they were also found together in colicins such as colicin E9 and E7, although sometimes they were separated by a receptor binding domain. Interestingly, a similar HNH nuclease-like domain was also found in effectors containing N-terminal domains such as LXG (5) and WXG100 (41), which are associated with T7SS, but without an accompanying Pyocin_S domain. A conservation logo generated by aligning the amino acid sequences of the VPA1263 C-terminal domain (737 to 869) and homologous domains found in known or predicted secreted toxins confirmed the presence of a conserved His-Gln-His motif (42) (Fig. S1B). These results led us to hypothesize that VPA1263 contains a C-terminal HNH nuclease domain that can mediate its antibacterial toxicity.

To determine the minimal region sufficient for VPA1263-mediated antibacterial toxicity, we ectopically expressed VPA1263 or its truncated versions (Fig. 1B) in *Escherichia coli* cells. As shown in Fig. 1C, the C-terminal end, corresponding to the predicted HNH nuclease domain (amino acids 737 to 869; HNH), was necessary and sufficient to intoxicate *E. coli* cells. Notably, the expression of all VPA1263 versions, except for the toxic HNH domain, was detected in immunoblots (Fig. S2A). Coexpression of the cognate immunity protein, Vti2 (3), rescued *E. coli* cells from the toxicity mediated by either the full-length VPA1263 or the C-terminal HNH nuclease domain (Fig. 1D), confirming that the toxicity mediated by the C-terminal HNH nuclease domain resulted from the same activity that is used by the full-length VPA1263. In addition, replacing the conserved histidine 837 with alanine (FL^HA or PyS-HNH^HA) abrogated VPA1263-mediated toxicity (Fig. 1E), further supporting the role of the HNH nuclease domain in VPA1263-mediated antibacterial toxicity. Notably, the expression of the mutated proteins was confirmed in immunoblots (Fig. S2B). Taken together, our results suggest that VPA1263 exerts its antibacterial toxicity via a C-terminal HNH nuclease domain.

**VPA1263 is a DNase.** Since HNH nucleases target DNA, we hypothesized that VPA1263 is a DNase. To test this hypothesis, we first set out to determine whether the C-terminal domain of VPA1263 could cleave DNA *in vitro* and *in vivo*. Since we were unable to purify the minimal toxin domain (amino acids 737 to 869), possibly due to low expression levels, we purified a truncated version encompassing both the C-terminal Pyocin_S and HNH nuclease domains (amino acids 547 to 869; PyS-HNH) (Fig. S3A). As expected, this toxic region was sufficient to cleave purified *E. coli* genomic DNA *in vitro* in the presence of $MgCl_2$, similarly to DNase I, which was used as a control (Fig. 2A). In contrast, a mutated version in which histidine 837 was replaced with alanine (PyS-HNH^HA) did not cleave the DNA. Furthermore, we were only able to isolate small amounts of genomic DNA from *E. coli* cultures expressing an arabinose-inducible PyS-HNH or the *Bacillus cereus* DNase, BC3021 (PoNe^BC [22]) (Fig. 2B), indicating that VPA1263 also cleaves DNA *in vivo*.

Next, we set out to determine whether VPA1263 targets prey DNA during T6SS-mediated bacterial competition. To this end, we used fluorescence microscopy to visualize the DNA in bacterial cultures after T6SS-mediated self-competition. From a *V. parahaemolyticus* prey strain that constitutively expresses a green fluorescent protein (GFP; used to distinguish prey from attacker cells), we deleted *vpa1263* and *vti2* (to specifically sensitize it to VPA1263-mediated toxicity) and competed the strain against attacker strains with a constitutively active T6SS1 deleted for the T6SS1 repressor *hns* to maximize T6SS1 activity

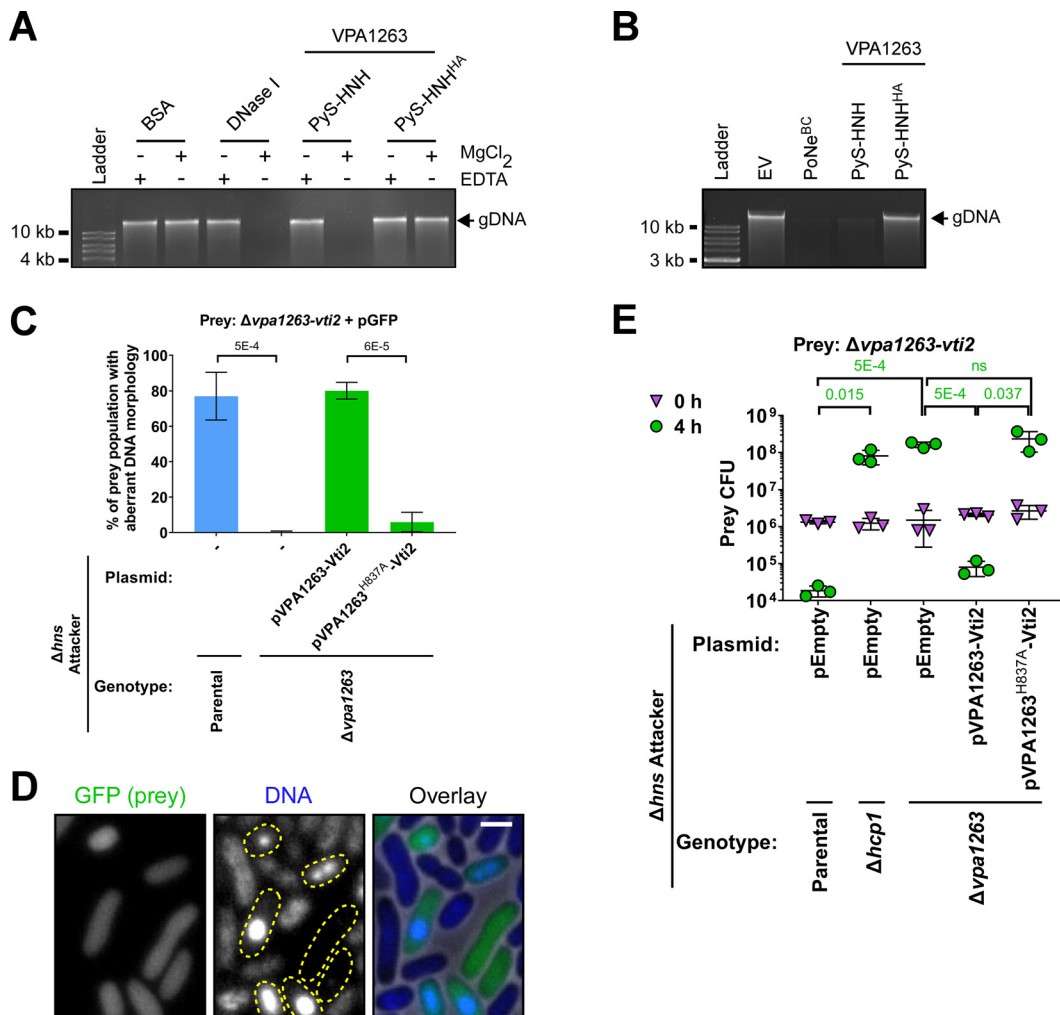

**FIG 2** VPA1263 is a DNase. (A) *In vitro* DNase activity. Purified *E. coli* genomic DNA (gDNA) was coincubated with the indicated purified protein in the presence (+) or absence (−) of MgCl$_2$ or EDTA for 5 min at 30°C. The results shown represent two independent experiments. BSA, bovine serum albumin. (B) *In vivo* DNase activity. The integrity of gDNA was determined after its isolation from *E. coli* MG1655 cells, in which the indicated proteins were expressed from arabinose-inducible expression plasmids. The results shown represent at least three independent experiments. EV, empty plasmid; PyS-HNH, amino acids 547 to 869 of VPA1263; PyS-HNH$^{HA}$, PyS-HNH with an H837A mutation; PoNe$^{BC}$, a PoNe domain-containing DNase from *Bacillus cereus* (BC3021). (C) Quantification of the percentage in the population of *V. parahaemolyticus* RIMD 2210633 Δ*vpa1263-vti2* prey cells showing aberrant DNA morphology (no detectable DNA or DNA foci) after 3 h of competition against the indicated attacker strains. DNA was visualized by Hoechst staining. The results represent the average ± SD of three independent experiments. Statistical significance between samples, determined using an unpaired two-tailed Student's *t* test, is denoted at the top of the panel. In each experiment, at least 100 prey cells were randomly selected and manually evaluated per treatment. (D) Sample fluorescence microscope images of prey cells after 3 h of competition against a Δ*hns* attacker strain, as described for panel C. Dashed yellow shapes in the DNA channel encircle prey cells detected in the GFP channel. Bar = 1 μm. (E) Viability counts of the indicated prey strain before (0 h) and after (4 h) coincubation with the indicated *V. parahaemolyticus* RIMD 2210633 Δ*hns* derivative attacker strain on MLB agar plates supplemented with L-arabinose at 30°C. Prey strains contain an empty plasmid that provides a selection marker. Data are shown as the mean ± SD; *n* = 3 technical replicates. Statistical significance between samples at the 4 h time point, determined using an unpaired two-tailed Student's *t* test, is denoted at the top of the panel. The experiment was performed three times with similar results; the results of a representative experiment are shown.

(43), with or without a deletion in *vpa1263* (Δ*hns*/Δ*vpa1263* and Δ*hns*, respectively). Notably, these attacker strains exhibited similar growth rates (Fig. S3B), and the Δ*hns*/Δ*vpa1263* strain retained the ability to intoxicate *Vibrio natriegens* prey in competition, indicating that it was able to deliver other T6SS1 effectors (3); thus, T6SS1 remained functional, even though this attacker strain was unable to intoxicate a Δ*vpa1263-vti2* prey strain (Fig. S3C). As shown in Fig. 2C and D and Fig. S4, VPA1263-sensitive, GFP-expressing prey cells devoid of DNA or containing DNA foci, which are probably regions of stress-

induced DNA condensation (44), were only detected after competition against the VPA1263$^+$ attacker strain ($\Delta hns$). Moreover, a plasmid for expression of VPA1263 and its cognate immunity protein, Vti2 (pVPA1263-Vti2), complemented this phenotype in a $\Delta hns/\Delta vpa1263$ attacker strain background, whereas a similar plasmid for expression of the catalytically inactive mutant, VPA1263$^{H837A}$ (pVPA1263$^{H837A}$-Vti2), did not (Fig. 2C). Taken together with the inability of a plasmid-encoded catalytically inactive mutant to complement the VPA1263-mediated toxicity of a $\Delta hns/\Delta vpa1263$ attacker strain during bacterial competition (Fig. 2E), these results support the conclusion that VPA1263 is a T6SS1 effector that exerts its toxicity during bacterial competition via its DNase activity.

**The MIX domain is necessary for T6SS-mediated secretion of VPA1263.** After determining that its toxicity is mediated by DNase activity, we next sought to investigate the role of VPA1263's MIX domain. Since MIX domains are found at the N termini of diverse polymorphic toxins that are secreted by T6SS (3), we hypothesized that MIX plays a role in T6SS-mediated secretion. To test this, we monitored the T6SS1-mediated secretion of C-terminal Myc-His-tagged VPA1263 variants expressed from a plasmid in a *V. parahaemolyticus* $\Delta hns$ strain with a constitutively active T6SS1 (T6SS1$^+$) or in a $\Delta hns/\Delta hcp1$ mutant with an inactive T6SS1 (T6SS1$^-$). Notably, to avoid self-intoxication of strains overexpressing VPA1263 variants containing the HNH nuclease domain, we used the catalytically inactive mutant H837A (Fig. 1E). The full-length protein (FL$^{HA}$) was secreted into the growth medium in a T6SS1-dependent manner, confirming previous comparative proteomics results (3) (Fig. 3A). An N-terminal region, including the MIX and LysM domains (N-ter), retained the ability to secrete via T6SS1, whereas the C-terminal region, containing the Pyocin_S and HNH nuclease toxin domains (PyS-HNH$^{HA}$), did not (Fig. 3A), indicating that the information required for T6SS1-mediated secretion is found in the N-terminal region. Remarkably, deletion of the region encoding amino acids 226 to 328 ($\Delta$MIX$^{HA}$), encompassing most of the MIX domain, resulted in the loss of T6SS1-mediated secretion. This result suggests that MIX is required for T6SS1-mediated secretion of VPA1263. In support of this conclusion, replacing residues belonging to the invariant GxxY motif in the MIX domain (3) with alanine, i.e., glycine 247 and tyrosine 250 (FL$^{GA/HA}$ and FL$^{YA/HA}$, respectively), also eliminated VPA1263's secretion via T6SS1. Notably, we validated that T6SS1 was functional in the strains expressing the VPA1263 variants that were not secreted by detecting the T6SS1-mediated secretion of the hallmark T6SS protein, VgrG1 (Fig. 3A). These results provide the first experimental indication that MIX is required for T6SS-mediated secretion of an effector. This conclusion was further supported by bacterial competition assays, in which VPA1263 with a mutation in the invariant glycine of the MIX domain (G247) was unable to complement the loss of prey intoxication by a $\Delta hns/\Delta vpa1263$ attacker (Fig. 3B). Attempts to determine whether the MIX domain is sufficient to mediate secretion via T6SS were inconclusive, due to the low expression level of VPA1263 truncations containing only the MIX domain region.

## DISCUSSION

In this work, we investigated the T6SS MIX effector VPA1263; we identified the roles of its C-terminal toxin domain and its MIX domain. Our results revealed that VPA1263 is a DNase toxin that requires its MIX domain for T6SS-mediated secretion.

Using a combination of toxicity assays, *in vivo* and *in vitro* biochemical assays, and fluorescence microscopy, we confirmed our computational prediction that VPA1263 has a C-terminal DNase domain that mediates its toxicity during bacterial competition. The identified toxin belongs to the widespread and diverse HNH nuclease superfamily (35–40); homologs are found in various known and predicted bacterial toxins. Interestingly, in some instances, this toxin domain is preceded by a Pyocin_S domain, as is the case in VPA1263. Pyocin_S was recently suggested to mediate the transport of DNase toxins across the inner membrane, from the periplasm to the cytoplasm (45). This activity was shown to be mediated by specific inner membrane proteins that serve as receptors. It will be interesting to determine whether VPA1263 and other Pyocin_S-

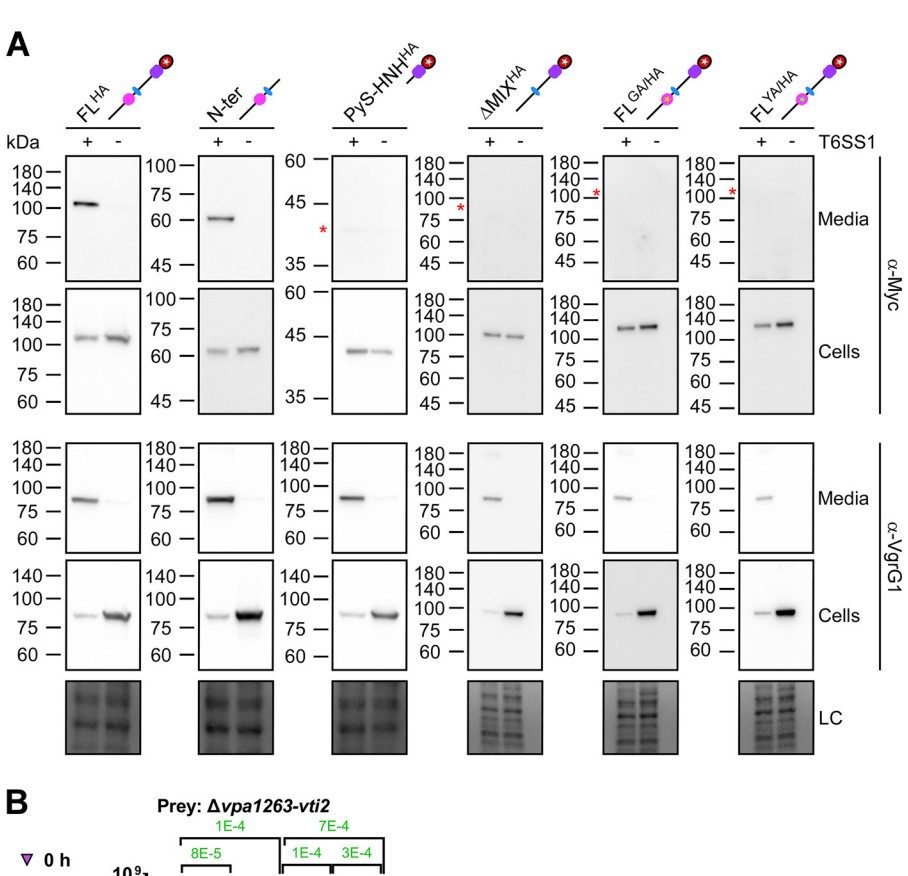

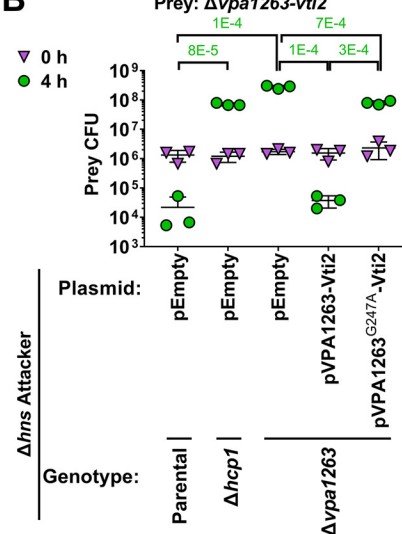

**FIG 3** MIX is required for secretion of VPA1263 via T6SS1. (A) Expression (cells) and secretion (media) of the indicated C-terminal Myc-His-tagged VPA1263 variants from *V. parahaemolyticus* RIMD 2210633 Δ*hns* (T6SS1⁺) or Δ*hns*Δ*hcp1* (T6SS1⁻). Proteins were expressed from an arabinose-inducible plasmid, and samples were resolved on SDS-PAGE. VPA1263 variants and the endogenous VgrG1 were visualized using specific antibodies against Myc and VgrG1, respectively. FL^HA, VPA1263 with an H387A mutation; N-ter, amino acids 1 to 546 of VPA1264; PyS-HNH, amino acids 547 to 869 of VPA1263; PyS-HNH^HA, PyS-HNH with a H837A mutation; FL^GA/HA, FL^HA with a G247A mutation; FL^YA/HA, FL^HA with a Y250A mutation. Schematic representations of the expressed VPA1263 variants are shown above; denoted domains and colors correspond to Fig. 1B; stars denote point mutations in the expressed variant. Red asterisks denote the expected size of the indicated proteins in the medium fractions. Loading control (LC) is shown for total protein lysates. (B) Viability counts of the indicated prey strain before (0 h) and after (4 h) coincubation with the indicated *V. parahaemolyticus* RIMD 2210633 Δ*hns* derivative attacker strains on MLB agar plates supplemented with ʟ-arabinose at 30°C. Prey strains contain an empty plasmid that provides a selection marker. Data are shown as the mean ± SD; *n* = 3 technical replicates. Statistical significance between samples at the 4 h time point, determined using an unpaired two-tailed Student's *t* test, is denoted at the top of the panel. The experiment was performed three times with similar results; the results of a representative experiment are shown.

containing T6SS effectors require this domain for transport into the cytoplasm to mediate toxicity, since it remains unclear whether T6SS effectors are delivered directly into the recipient cell cytoplasm, periplasm, or randomly into either compartment (46–48). Notably, we recently reported that VPA1263 selectively intoxicates bacterial strains when delivered via an engineered T6SS in *V. natriegens* (49). VPA1263-delivering attacker strains were toxic to *Vibrio* and *Aeromonas* strains but had no effect on the viability of *E. coli* or *Salmonella* prey (49). It is therefore tempting to speculate that differences in the potential inner membrane receptors of the VPA1263 Pyocin_S domain are responsible for the observed selective toxicity. If true, it may represent a previously unappreciated mode of natural resistance against T6SS effectors.

Importantly, we found that MIX is required for T6SS-mediated secretion of VPA1263; even single point mutations in the invariant GxxY motif were sufficient to abolish effector secretion and effector-mediated intoxication of sensitive recipient prey bacteria during competition. While this is the first experimental evidence of a role for MIX in secretion, the mechanism by which it contributes to secretion remains unknown. It is possible that MIX contributes toward a stable or desirable effector conformation that is required for correct loading or positioning on the T6SS. Alternatively, MIX may mediate interaction with a secreted tube or spike component. We will investigate the underlying mechanism by which MIX mediates T6SS secretion in future work.

## MATERIALS AND METHODS

**Strains and media.** For a complete list of the strains used in this study, see Table S1 in the supplemental material. *Escherichia coli* strains were grown in 2xYT broth (1.6% [wt/vol] tryptone, 1% [wt/vol] yeast extract, and 0.5% [wt/vol] NaCl) or on lysogeny broth (LB) agar plates containing 1% (wt/vol) NaCl at 37°C, or at 30°C when harboring effector expression plasmids. The media were supplemented with chloramphenicol (10 $\mu$g/mL) or kanamycin (30 $\mu$g/mL) to maintain plasmids and with 0.4% (wt/vol) glucose to repress protein expression from the arabinose-inducible promoter, P*bad*. To induce expression from P*bad*, L-arabinose was added to media at 0.05 to 0.1% (wt/vol), as indicated.

*Vibrio parahaemolyticus* RIMD 2210633 (50) and its derivative strains, as well as *V. natriegens* ATCC 14048, were grown in MLB medium (LB medium containing 3% [wt/vol] NaCl) or on marine minimal medium (MMM) agar plates (1.5% [wt/vol] agar, 2% [wt/vol] NaCl, 0.4% [wt/vol] galactose, 5 mM MgSO$_4$, 7 mM K$_2$SO$_4$, 77 mM K$_2$HPO$_4$, 35 mM KH$_2$PO$_4$, and 2 mM NH$_4$Cl) at 30°C. When the vibrios contained a plasmid, the media were supplemented with kanamycin (250 $\mu$g/mL) or chloramphenicol (10 $\mu$g/mL) to maintain the plasmid. To induce expression from P*bad*, L-arabinose was added to the media at 0.05% (wt/vol).

**Plasmid construction.** For a complete list of plasmids used in this study, see Table S2. For protein expression, the coding sequences (CDS) of *vpa1263* (GenBank protein accession number BAC62606.1) and *vti2* (BA000032.2; chromosome 2, positions 1344453 to 1344737; this gene encodes a protein identical to GenBank accession number WP_005477334.1) were amplified from genomic DNA of *V. parahaemolyticus* RIMD 2210633. The amplification products were inserted into the multiple cloning site (MCS) of pBAD$^K$/Myc-His or pBAD33.1$^F$ using the Gibson assembly method (51). The plasmids were introduced into *E. coli* cells using electroporation. Transformants were grown on agar plates supplemented with 0.4% (wt/vol) glucose to repress unwanted expression from the P*bad* promoter during the subcloning steps. The plasmids were introduced into *V. parahaemolyticus* cells via conjugation. The transconjugants were grown on MMM agar plates supplemented with appropriate antibiotics to maintain the plasmids.

**Construction of deletion strains.** The construction of pDM4-based (52) plasmids for deletion of *vpa1263*, *vpa1263-vti2*, and *hns* (*vp1133*) was reported previously (3, 43). In-frame deletions of *vpa1263*, *vpa1263-vti2*, and *hns* in *V. parahaemolyticus* RIMD 2210633 were performed as previously described (53). Briefly, a Cm$^R$OriR6K suicide plasmid, pDM4, containing ~1 kb upstream and ~1 kb downstream of the gene to be deleted in its MCS was conjugated into *V. parahaemolyticus* cells, and transconjugants were selected on solid medium plates supplemented with chloramphenicol. Then, bacteria were counterselected on solid medium plates containing 15% (wt/vol) sucrose for loss of the *sacB*-containing plasmid. Deletions were confirmed by PCR.

**Toxicity in *E. coli*.** Bacterial toxicity assays were performed as previously described (21), with minor changes. Briefly, to assess the toxicity mediated by VPA1263 or its truncated versions, pBAD$^K$/Myc-His plasmids encoding the indicated proteins were transformed into *E. coli* MG1655. The *E. coli* transformants were streaked onto either repressing (containing 0.4% [wt/vol] glucose) or inducing (containing 0.05% [wt/vol] L-arabinose) LB agar plates supplemented with kanamycin. Chloramphenicol was also included in the medium when a pBAD33.1$^F$-based plasmid was used. The plates were incubated at 30°C for 16 h and then imaged using a Fusion FX6 imaging system (Vilber Lourmat). The experiments were performed at least three times, with similar results.

**Protein expression in *E. coli*.** Protein expression in *E. coli* was performed as previously described (21), with minor changes. Briefly, to assess the expression of C-terminal Myc-tagged proteins encoded on arabinose-inducible plasmids, overnight cultures of *E. coli* MG1655 containing the indicated plasmids were washed twice with fresh 2xYT broth to remove residual glucose. Following normalization to an

optical density at 600 nm ($OD_{600}$) of 0.5 in 3 mL of 2xYT broth supplemented with appropriate antibiotics, the cultures were grown for 2 h at 37°C. To induce protein expression, 0.05% (wt/vol) L-arabinose was added to the media. After having grown for 2 additional hours at 37°C, 0.5-$OD_{600}$ units of cells were pelleted and resuspended in 2× Tris-glycine SDS sample buffer (Novex, Life Sciences). The samples were boiled for 5 min, and cell lysates were resolved on Mini-PROTEAN or Criterion TGX stain-free precast gels (Bio-Rad). For immunoblotting, $\alpha$-Myc antibodies (Santa Cruz Biotechnology; 9E10; mouse monoclonal antibody (MAb); sc-40) were used at a dilution of 1:1,000. Protein signals were visualized in a Fusion FX6 imaging system (Vilber Lourmat) using enhanced chemiluminescence (ECL) reagents.

**Protein purification.** To purify truncated VPA1263 proteins for the *in vitro* DNase assays, *E. coli* BL21 (DE3) cells harboring plasmids for arabinose-inducible expression of the indicated Myc-His-tagged VPA1263 variants and the FLAG-tagged Vti2 (the immunity protein required to antagonize the toxicity of VPA1263 variants inside bacteria) were grown for 16 h in 2xYT medium supplemented with kanamycin and chloramphenicol at 37°C. The bacterial cultures were then diluted 100-fold into fresh medium and incubated at 37°C with agitation (180 rpm). When the bacterial cultures reached an $OD_{600}$ of ~1.0, L-arabinose was added to the medium (to a final concentration of 0.1% [wt/vol]) to induce protein expression, and the cultures were grown for 4 additional hours at 30°C. Cells were harvested by centrifugation at 4°C (20 min, at 13,300 × $g$), followed by washing with a 0.9% (wt/vol) NaCl solution to remove residual medium. Then, cells were resuspended in 3 mL lysis buffer A (20 mM Tris-Cl [pH 7.5], 500 mM NaCl, 5% [vol/vol] glycerol, 10 mM imidazole, 0.1 mM phenylmethylsulfonyl fluoride [PMSF], and 8 M urea). Urea was included in the buffer to denature the proteins and thus release Vti2 from the VPA1263 variants. The cells were disrupted using a high-pressure cell disruptor (Constant Systems One Shot cell disruptor; model code, MC/AA). To remove cell debris, the lysates were centrifuged for 20 min at 13,300 × $g$ and 4°C. Next, 250 $\mu$L Ni-Sepharose resin (50% slurry; GE Healthcare) was prewashed with lysis buffer A and then mixed with the supernatant fractions of lysed cells containing the denatured proteins. The suspensions were incubated for 1 h at 4°C with constant rotation and then loaded onto a column. The immobilized resin was washed with 10 mL wash buffer A (20 mM Tris-Cl [pH 7.5], 500 mM NaCl, 5% [vol/vol] glycerol, 40 mM imidazole, and 8 M urea). Bound proteins were eluted from the column using 1 mL elution buffer A (20 mM Tris-Cl [pH 7.5], 500 mM NaCl, 5% [vol/vol] glycerol, 500 mM imidazole, and 8 M urea). The presence and purity of the eluted Myc-His-tagged VPA1263 variants were confirmed using SDS-PAGE, using stain-free gels (Bio-Rad).

To refold the denatured, purified proteins before *in vitro* DNase activity assays, a refolding procedure was applied. The eluted proteins were dialyzed against DNase assay buffer (20 mM Tris-Cl [pH 7.5], 200 mM NaCl, and 5% [vol/vol] glycerol) and incubated for an hour at 4°C. The buffer was replaced twice to remove unwanted imidazole and urea. Then, the 1-mL suspension was concentrated to ~300 $\mu$L using a Spin-X ultrafiltration (UF) concentrator column (Corning; 30 kDa). The purified proteins were quantified using the Bradford method with 5× Bradford reagent (Bio-Rad). The procedure was carried out at 4°C.

***In vitro* DNase assays.** For determining the *in vitro* DNase activity, genomic DNA isolated from *E. coli* BL21(DE3) (200 ng) was incubated with 0.5 $\mu$g of purified VPA1263 variants for 5 min at 30°C in DNase assay buffer (20 mM Tris-Cl [pH 7.5], 200 mM NaCl, and 5% [vol/vol] glycerol) supplemented with either 2 mM EDTA (a metal chelator) or $MgCl_2$. The total volume of the reactions was 20 $\mu$L. The reactions were stopped by adding 6.65 $\mu$g of proteinase K, and the samples were incubated for 5 min at 55°C. The samples were analyzed by 1.0% agarose-gel electrophoresis. For the positive and negative controls, 1 U DNase I (Thermo Fisher Scientific) and 0.5 $\mu$g bovine serum albumin (BSA) (Sigma) were used, respectively. The experiments were performed twice, with similar results.

***In vivo* DNase assays.** *E. coli* MG1655 strains containing the indicated pBAD$^K$/Myc-His plasmid, either empty or encoding VPA1263$^{547-869}$ (PyS-HNH), VPA1263$^{547-869/H837A}$ (PyS-HNH$^{HA}$), or PoNe$^{Bc}$ (BC3021), were grown overnight in 2xYT medium supplemented with kanamycin and 0.4% (wt/vol) glucose. The overnight cultures were washed with 2xYT medium and normalized to an $OD_{600}$ of 1.0 in 3 mL of fresh 2xYT supplemented with kanamycin and 0.1% (wt/vol) L-arabinose (to induce protein expression). The cultures were grown for 90 min with agitation (220 rpm) at 37°C before 1.0 $OD_{600}$ units were pelleted. Genomic DNA was isolated from each sample using the EZ spin column genomic DNA kit (Bio Basic) and eluted with 30 $\mu$L ultrapure water (Milli-Q). Equal genomic DNA elution volumes were analyzed by 1.0% (wt/vol) agarose gel electrophoresis. The experiments were performed at least three times, with similar results.

***Vibrio* growth assays.** *Vibrio* growth assays were performed as previously described (21). Briefly, overnight cultures of *V. parahaemolyticus* strains were normalized to $OD_{600}$ = 0.01 in MLB and transferred to 96-well plates (200 $\mu$L per well; $n$ = 3 technical replicates). The 96-well plates were incubated in a microplate reader (BioTek SYNERGY H1) at 30°C with constant shaking at 205 cpm. $OD_{600}$ reads were acquired every 10 min. The experiments were performed at least three times, with similar results.

**Bacterial competition assays.** Bacterial competition assays were performed as previously described (21). Briefly, attacker and prey strains were grown overnight in MLB with the addition of antibiotics when maintenance of plasmids was required. The bacterial cultures were normalized to $OD_{600}$ = 0.5 and mixed at a 4:1 attacker:prey ratio. The mixtures were spotted onto MLB agar plates and incubated for 4 h at 30°C. The plates were supplemented with 0.1% (wt/vol) L-arabinose when expression from an arabinose-inducible plasmid in the attacker strain was required. The number of CFU of the prey strain was determined by growing the mixtures on selective plates at the 0 and 4-h time points. The experiments were performed at least three times, with similar results.

**Fluorescence microscopy.** Assessment of prey DNA morphology during bacterial competition was performed as previously described (22). Briefly, the indicated *V. parahaemolyticus* RIMD 2210633 strains

were grown overnight and mixed as described above for the bacterial competition assays. The Δ*vpa1263-vti2* prey strain constitutively expressed GFP (from a stable, high-copy-number plasmid [54]) to distinguish them from attacker cells. The attacker-prey mixtures were spotted onto MLB agar plates and incubated at 30°C for 3 h. The plates were supplemented with 0.1% (wt/vol) L-arabinose when expression from an arabinose-inducible plasmid in the attacker strain was required. Bacteria were then scraped from the plates, washed with M9 medium, and incubated at room temperature for 10 min in M9 medium containing Hoechst 33342 (Invitrogen) DNA dye at a final concentration of 1 $\mu$g/$\mu$L. The cells were then washed and resuspended in 30 $\mu$L M9 medium. One microliter of bacterial suspension was spotted onto M9 agar (1.5% [wt/vol]) pads and allowed to dry for 2 min before the pads were placed face down into 35 mm glass bottom CELLview cell culture dishes. Bacteria were imaged using a Nikon Eclipse Ti2-E inverted motorized microscope equipped with a CFI PLAN apochromat DM 100× oil lambda PH-3 (NA, 1.45) lens objective, a Lumencor SOLA SE II 395 light source, and ET-DAPI (number 49028; used to visualize the Hoechst signals) and ET-EGFP (number 49002; used to visualize the GFP signals) filter sets. Images were acquired using a DS-Qi2 Mono cooled digital microscope camera (16 megapixel [MP]) and were postprocessed using the Fiji ImageJ suite. Cells exhibiting a normal DNA morphology or aberrant DNA morphology (i.e., DNA foci or no DNA) were manually counted (>100 GFP-expressing prey cells per treatment in each experiment). The experiment was repeated three times.

**Protein secretion assays.** *V. parahaemolyticus* RIMD 2210633 strains were grown overnight at 30°C in MLB supplemented with kanamycin to maintain plasmids. Bacterial cultures were normalized to an OD$_{600}$ value of 0.18 in 5 mL MLB supplemented with kanamycin and L-arabinose (0.05% [wt/vol]) to induce expression from P*bad* promoters. After 4 h of incubation at 30°C with agitation (220 rpm), 1.0 OD$_{600}$ units of cells were collected for expression fractions (cells). The cell pellets were resuspended in 2× Tris-glycine SDS sample buffer (Novex, Life Sciences). For secretion fractions (media), suspension volumes equivalent to 10 OD$_{600}$ units of cells were filtered (0.22 $\mu$m), and proteins were precipitated from the media using deoxycholate and trichloroacetic acid (55). Cold acetone was used to wash the protein precipitates twice. Then, the protein precipitates were resuspended in 20 $\mu$L of 10 mM Tris-HCl (pH 8), followed by the addition of 20 $\mu$L of 2× Tris-glycine SDS sample buffer supplemented with 5% $\beta$-mercaptoethanol and 0.5 $\mu$L 1 N NaOH to maintain a basic pH. The expression and secretion samples were boiled for 5 min and then resolved on Mini-PROTEAN or Criterion TGX stain-free precast gels (Bio-Rad). For immunoblotting, primary antibodies were used at a concentration of 1:1,000. The following antibodies were used: $\alpha$-Myc antibodies (Santa Cruz Biotechnology; 9E10; mouse MAb; sc-40) and custom-made $\alpha$-VgrG1 (56). Protein signals were visualized in a Fusion FX6 imaging system (Vilber Lourmat) using enhanced chemiluminescence (ECL) reagents.

**VPA1263 domain and homolog analysis.** Domains in VPA1263 (GenBank protein accession number BAC62606.1) were identified using the NCBI conserved domain database (31) or by homology detection and structure prediction using a hidden Markov model (HMM)-HMM comparison, i.e., the HHpred tool (34). Proteins containing a specialized secretion system delivery domain or known secreted toxins bearing homology to the C-terminal HNH nuclease toxin domain of VPA1263 were identified through an iterative search against the UniProt protein sequence database using Jackhmmer (57). Homologs containing diverse secretion-associated domains (e.g., domains found in T6SS or T7SS polymorphic toxins) were selected, and their C-terminal ends were aligned using MEGA X (58) with MUSCLE (59); alignment columns not represented in VPA1263 were removed. Conserved residues were illustrated using the WebLogo v3 server (60). Amino acid numbering was based on the sequence of VPA1263.

## SUPPLEMENTAL MATERIAL

Supplemental material is available online only.
**SUPPLEMENTAL FILE 1**, PDF file, 0.8 MB.

## ACKNOWLEDGMENTS

We thank members of the Salomon lab for helpful discussions and suggestions and Dan Goldenberg for technical assistance.

This project received funding from the European Research Council, under the European Union's Horizon 2020 Research and innovation program (grant agreement number 714224 to D. Salomon), and from the Israel Science Foundation (grant number 920/17 to D. Salomon and 1362/21 to D. Salomon and E. Bosis). C. M. Fridman was supported by scholarships from the Clore Israel Foundation and from the Manna Center Program in Food Safety and Security at Tel Aviv University, as well as by a scholarship for outstanding doctoral students from the Orthodox community from the Council for Higher Education.

The funders had no role in study design, data collection and interpretation, or the decision to submit the work for publication. This work was performed in partial fulfillment of the requirements for a Ph.D. degree for C. M. Fridman at the Sackler Faculty of Medicine, Tel Aviv University.

C. M. Fridman: Conceptualization, Investigation, Methodology, and Writing – Original Draft. B. Jana: Conceptualization, Investigation, Methodology, and Writing – Review and Editing. R. Ben-Yaakov: Investigation and Methodology. E. Bosis:

Conceptualization, Investigation, Methodology, Funding Acquisition, and Writing – Review and Editing. D. Salomon: Conceptualization, Supervision, Funding Acquisition, Investigation, Methodology, and Writing – Original Draft.

We declare that we have no conflicts of interest.

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
