## [Reviewer comments · Microbiology Spectrum]

Microbiology Spectrum

A DNase T6SS effector requires its MIX domain for secretion

Chaya Fridman, Biswanath Jana, Rotem Ben-Yaakov, Eran Bosis, and Dor Salomon

Corresponding Author(s): Dor Salomon, Tel Aviv University

Review Timeline:

Submission Date:	June 29, 2022
Editorial Decision:	July 25, 2022
Revision Received:	August 17, 2022
Accepted:	August 26, 2022

Editor: Ethel Bayer-Santos

Reviewer(s): Disclosure of reviewer identity is with reference to reviewer comments included in decision letter(s). The following individuals involved in review of your submission have agreed to reveal their identity: Erh-Min Lai (Reviewer #2)

Transaction Report:

DOI: <https://doi.org/10.1128/spectrum.02465-22>

July 25, 2022

Dr. Dor Salomon
Tel Aviv University
Clinical Microbiology and Immunology
Tel Aviv
Israel

Re: Spectrum02465-22 (A DNase T6SS effector requires its MIX domain for secretion)

Dear Dr. Dor Salomon:

Thank you for submitting your manuscript to Microbiology Spectrum. Your manuscript has been reviewed by two experts in the field who provided constructive comments to improve your work. Please read them carefully and when submitting the revised version of your paper, please provide (1) point-by-point responses to the issues raised by the reviewers as file type "Response to Reviewers," not in your cover letter, and (2) a PDF file that indicates the changes from the original submission (by highlighting or underlining the changes) as file type "Marked Up Manuscript - For Review Only". Please use this link to submit your revised manuscript - we strongly recommend that you submit your paper within the next 60 days or reach out to me. Detailed instructions on submitting your revised paper are below.

Link Not Available

Sincerely,

Ethel Bayer-Santos
Editor, Microbiology Spectrum

Journals Department
Reviewer comments:

Reviewer #1 (Comments for the Author):

In this manuscript Fridman, Jana and colleagues describe a multidomain effector VPA1263 secreted by T6SS of *Vibrio parahaemolyticus*. This polymorphic toxin contains the MIX, LysM, Pyosin_S and HNH domains. Authors demonstrate in vivo and in vitro that the toxicity comes from HNH domain that functions as a DNase. They also show for the first time that the N terminal MIX domain is required for secretion of the VPA1263 effector, but not for the toxicity. In the last part, which seems to be less connected to the rest of the story, they investigate the genetic context of this effector to look for similar genetic islands and conclude that they carry various defense weapons - antibacterial, antiphage or both. The manuscript is well written and clear, although a few details could be added to better understand the context. Overall, I think the manuscript demonstrates some important details about multidomain effectors, especially MIX domain and more generally T6SS of *Vibrio*'s and that it would be interesting for the field.

Figure 1 - for clarity I would suggest to label the constructs in C, D and E by the domains that they contain, for example it could be "HNH" for "737-869", "Pyocin_S-HNH" for "547-869", and so on. Those names can be additionally provided on the left side next to the actual numbers in B panel. I also doubt that the scheme of plate is useful, it could simply be annotated outside of the "inducing" plate, which is where we ultimately look. Currently one has to go back-and-forth between the different parts to figure out who is what and gives which result.

Figure 2 - consider also naming the 547-869 as HNH domain

Line 142 - I suggest clarifying here that GFP was used to distinguish prey cells, because at first I had doubts whether it is a reporter for viability or translation and I found this more clear in methods.

Line 144 - consider adding one sentence to explain that the constitutively active T6SS is the hns mutant, which I have guessed from the use of the strain in next sentence. Later this information appeared in line 162, but it should be clear here, at the first mention.

Line 148 - consider adding short explanation why T6SS1 remained functional? Does it secrete other toxins?

Line 150 - what do these „DNA loci" signify? I suppose it could be genome condensation which was previously described as an outcome of double-strand breaks. It's a pity that authors do not comment on this phenomenon at all, while it captures attention in figures 2C and D.

Line 188 - I find it intriguing that the Septu systems also contain HNH domain. I think that one sentence on what we know on Septu would be nice, even if there's little known.

Line 192 - while transposase, integrase, helicase are common, I have never heard the GmtX and GmtY and had to look it up. Even if I found very little information, it should still be explained what they are or that they are suggested to be, especially that later they are referred as „mobility genes".

Line 236 - did the authors consider making Pyocin_S domain mutant and testing the killing against the vpa1263-vti2 + gfp strain? (just a suggestion, not a requirement for this manuscript)

Line 256 - should you briefly explain the „stuffing model" here? I am not sure I could follow the hypothesis.

Line 274 - shouldn't it be plural „incomplete Vibrio genomes"?

Line 275 - what about variant 7? I also don't see defense weapons, but there are hypothetical ORFs.

Reviewer #2 (Comments for the Author):

Summary:

This manuscript reported the biochemical function of a T6SS effector, VPA1263 encoded in the VPAl-6 genomic island from *Vibrio parahaemolyticus*, and the domain/motif required for its secretion. Previous study found that VPA1263 as an antibacterial toxin belonging to a widespread class of T6SS polymorphic effectors containing a MIX domain. In this work, the authors showed that the C-terminal toxin domain is a DNase belonging to the HNH nuclease family and the conserved Histidine of HNH motif is required for DNase activity shown by in vitro and in vivo assays. The author further showed that N-terminal MIX domain is not required for VPA1263 in exhibiting toxicity when expressed in *E. coli* or DNase activity but required for secretion of VPA1263 from *V. parahaemolyticus*. The author also provide evidence that the conserved GxxY motif in MIX domain is required for secretion. By genome wide survey, they further showed that VPAl-6 genomic islands carry mobile genetic elements and are present in various vibrios, which encode diverse antibacterial and anti-phage weapons. Overall, the manuscript reported findings that advance the knowledge of effector function, secretion, and its evolutionary perspectives. The manuscript is well written and the conclusions are generally supported with solid data. Below are some comments for further improvement of the manuscript.

Major comments:

1. HNH nuclease is a widespread family of DNase toxin found in T6SS. Thus, it is not new for reporting this family of DNase toxin. However, the authors did a good job in characterizing the DNase activity using both in vitro and in vivo assays by ectopic expression of various constructs and catalytic site variants. Thus, the evidence to claim the VPA1263 is a DNase, which exhibits its toxicity in *E. coli* is well documented. It is less convincing is whether its DNase activity is indeed responsible for exerting antibacterial activity during the interbacterial competition context. As shown in Figure 2C,D and Figure S3C, the data demonstrated the deletion of vpa1263 caused the loss of antibacterial activity to *V. parahaemolyticus* delta-vp1263-vti2 prey and reduced or condensed DNA signals in the prey cell. However, the evidence to conclude that VPA1263 uses its DNase activity for interbacterial competition is lacking. Thus, additional experiments using the attacker carrying the DNase inactive mutant for interbacterial competition and microscope observation are key experiments to claim this notion.
2. The finding of the N-terminal region including MIX domain is necessary and required for secretion is an exciting finding

although the underlying mechanism is yet to be investigated. It is understandable for the reason in using DNase inactive variant for secretion assay. However, interbacterial competition assay of the MIX domain mutants carrying WT DNase domain should be performed to validate their role in secretion and interbacterial competition. This is important as such evidence is required to warrant the findings with biological significance. Co-expression of immunity protein with each VPA1263 variants should be feasible for both secretion and interbacterial competition assays.

Minor comments:

1. Line 33-34, I would suggest to rephrase the statement "for the first time, that a domain used as a marker for secreted toxins is actually required for toxin secretion". In fact, I would have expected this requirement to be discovered when MIX domain was identified. Emphasis on this point seems unnecessary.
2. Data in Figure 1C showing the toxicity of various VPA1263 in E. coli is quite clear. I am curious about what is the degree of the toxicity? Although it is not essential, it would be more informative to carry out growth curve analysis to compare the degree of toxicity and how well immunity protein rescue the toxicity.
3. It is interesting that HND nuclease domain itself is not stable. I am also curious whether H837A mutation in HNH domain can stabilize this protein or this domain requires pyocin S domain for stability. The authors may test this idea or discuss this possibility.

Staff Comments:

Preparing Revision Guidelines

Please return the manuscript within 60 days; if you cannot complete the modification within this time period, please contact me. If you do not wish to modify the manuscript and prefer to submit it to another journal, please notify me of your decision immediately so that the manuscript may be formally withdrawn from consideration by Microbiology Spectrum.

Point-by-point reply to reviewers' comments.

General comment by authors:

We would like to thank the three reviewers for providing constructive and helpful comments that allowed us to improve our manuscript. The reviewers commented that the last part of the manuscript, describing the genome neighborhood of vpa1263, is preliminary and not well connected with the rest of the manuscript. We agree with this criticism. Since we realized that performing a more in-depth analysis looking at similar mobile elements outside the Vibrio family will result in two distinct stories, we decided to remove this part from the current report; we will continue to investigate it as part of a separate project.

Reviewer #1 (Comments for the Author):

In this manuscript Fridman, Jana and colleagues describe a multidomain effector VPA1263 secreted by T6SS of *Vibrio parahaemolyticus*. This polymorphic toxin contains the MIX, LysM, Pyosin_S and HNH domains. Authors demonstrate in vivo and in vitro that the toxicity comes from HNH domain that functions as a DNase. They also show for the first time that the N terminal MIX domain is required for secretion of the VPA1263 effector, but not for the toxicity. In the last part, which seems to be less connected to the rest of the story, they investigate the genetic context of this effector to look for similar genetic islands and conclude that they carry various defense weapons - antibacterial, antiphage or both. The manuscript is well written and clear, although a few details could be added to better understand the context. Overall, I think the manuscript demonstrates some important details about multidomain effectors, especially MIX domain and more generally T6SS of *Vibrio*'s and that it would be interesting for the field.

We thank the reviewer for the kind words and helpful suggestions.

Figure 1 - for clarity I would suggest to label the constructs in C, D and E by the domains that they contain, for example it could be "HNH" for "737-869", "Pyocin_S-HNH" for "547-869", and so on. Those names can be additionally provided on the left side next to the actual numbers in B panel. I also doubt that the scheme of plate is useful, it could simply be annotated outside of the "inducing" plate, which is where we ultimately look. Currently one has to go back-and-forth between the different parts to figure out who is what and gives which result.

The Figure was modified as suggested. The following Figures were also modified to maintain the same nomenclature.

Figure 2 - consider also naming the 547-869 as HNH domain

The names were modified to match the revised Fig. 1.

Line 142 - I suggest clarifying here that GFP was used to distinguish prey cells, because at first

I had doubts whether it is a reporter for viability or translation and I found this more clear in methods.

A clarification was added to the text, which now reads: "We competed a V. parahaemolyticus prey strain that constitutively expresses a green fluorescent protein (GFP; used to distinguish prey from attacker cells)..."

Line 144 - consider adding one sentence to explain that the constitutively active T6SS is the hns mutant, which I have guessed from the use of the strain in next sentence. Later this information appeared in line 162, but it should be clear here, at the first mention.

A clarification was added to the text, which now reads: "... against attacker strains with a constitutively active T6SS1 deleted for the T6SS1 repressor hns to maximize T6SS1 activity (43), with or without a deletion in vpa1263 (Δ hns/ Δ vpa1263 and Δ hns, respectively)."

Line 148 - consider adding short explanation why T6SS1 remained functional? Does it secrete other toxins?

A clarification was added to the text, which now reads: "... the Δ hns/ Δ vpa1263 strain retained the ability to intoxicate Vibrio natriegens prey in competition, indicating that it was able to deliver other T6SS1 effectors; thus, T6SS1 remained functional even though this attacker strain was unable to intoxicate a Δ vpa1263-vti2 prey strain..."

Line 150 - what do these „DNA loci" signify? I suppose it could be genome condensation which was previously described as an outcome of double-strand breaks. It's a pity that authors do not comment on this phenomenon at all, while it captures attention in figures 2C and D.

A clarification was added to the text, which now reads: "... VPA1263-sensitive, GFP-expressing prey cells devoid of DNA or containing DNA foci, which are probably regions of stress-induced DNA condensation (44), were only detected after competition against the VPA1263⁺ attacker strain (Δ hns)."

Line 188 - I find it intriguing that the Septu systems also contain HNH domain. I think that one sentence on what we know on Septu would be nice, even if there's little known.

As noted above, this section was removed from the manuscript.

Line 192 - while transposase, integrase, helicase are common, I have never heard the GmtX and GmtY and had to look it up. Even if I found very little information, it should still be explained what they are or that they are suggested to be, especially that later they are referred as „mobility genes".

As noted above, this section was removed from the manuscript.

Line 236 - did the authors consider making Pyocin_S domain mutant and testing the killing against the vpa1263-vti2 + gfp strain? (just a suggestion, not a requirement for this manuscript)

The role of the Pyocin_S domain is indeed intriguing, but we did not address it in the current study.

Line 256 - should you briefly explain the „stuffing model" here? I am not sure I could follow the hypothesis.

For simplicity, we decided to remove this part of the discussion. It now reads: "It is possible that MIX contributes toward a stable or desirable effector conformation that is required for correct loading or positioning on the T6SS. Alternatively, MIX may mediate interaction with a secreted tube or spike component. We will investigate the underlying mechanism by which MIX mediates T6SS secretion in future work."

Line 274 - shouldn't it be plural „incomplete Vibrio genomes"?

The reviewer is correct. Nevertheless, as noted above, this section was removed from the manuscript.

Line 275 - what about variant 7? I also don't see defense weapons, but there are hypothetical ORFs.

As noted above, this section was removed from the manuscript.

Reviewer #2 (Comments for the Author):

Summary:

This manuscript reported the biochemical function of a T6SS effector, VPA1263 encoded in the VPal-6 genomic island from *Vibrio parahaemolyticus*, and the domain/motif required for its secretion. Previous study found that VPA1263 as an antibacterial toxin belonging to a widespread class of T6SS polymorphic effectors containing a MIX domain. In this work, the authors showed that the C-terminal toxin domain is a DNase belonging to the HNH nuclease family and the conserved Histidine of HNH motif is required for DNase activity shown by in vitro and in vivo assays. The author further showed that N-terminal MIX domain is not required for VPA1263 in exhibiting toxicity when expressed in *E. coli* or DNase activity but required for secretion of VPA1263 from *V. parahaemolyticus*. The author also provide evidence that the conserved GxxY motif in MIX domain is required for secretion. By genome wide survey, they further showed that VPal-6 genomic islands carry mobile genetic elements and are present in various vibrios, which encode diverse antibacterial and anti-phage weapons. Overall, the manuscript reported findings that advance the knowledge of effector function, secretion, and its evolutionary perspectives. The manuscript is well written and the conclusions are generally supported with solid data. Below are some comments for further improvement of the manuscript.

We thank the reviewer for the helpful suggestions to improve our manuscript.

Major comments:

1. HNH nuclease is a widespread family of DNase toxin found in T6SS. Thus, it is not new for reporting this family of DNase toxin. However, the authors did a good job in characterizing the DNase activity using both in vitro and in vivo assays by ectopic expression of various constructs and catalytic site variants. Thus, the evidence to claim the VPA1263 is a DNase, which exhibits its toxicity in *E. coli* is well documented. It is less convincing is whether its DNase activity is indeed responsible for exerting antibacterial activity during the interbacterial competition context. As shown in Figure 2C,D and Figure S3C, the data demonstrated the deletion of *vpa1263* caused the loss of antibacterial activity to *V. parahaemolyticus* delta-*vp1263-vti2* prey and reduced or condensed DNA signals in the prey cell. However, the evidence to conclude that VPA1263 uses its DNase activity for interbacterial competition is lacking. Thus, additional experiments using the attacker carrying the DNase inactive mutant for interbacterial competition and microscope observation are key experiments to claim this notion.

As suggested, we now show that a mutation in the HNH active site (H837A) impairs VPA1263-mediated intoxication of a VPA1263-sensitive prey strain during competition (Fig. 2E). We also show that the same mutation abolishes the aberrant DNA morphologies observed in sensitive prey during competition (Fig. 2C).

2. The finding of the N-terminal region including MIX domain is necessary and required for secretion is an exciting finding although the underlying mechanism is yet to be investigated. It is

understandable for the reason in using DNase inactive variant for secretion assay. However, interbacterial competition assay of the MIX domain mutants carrying WT DNase domain should be performed to validate their role in secretion and interbacterial competition. This is important as such evidence is required to warrant the findings with biological significance. Co-expression of immunity protein with each VPA1263 variants should be feasible for both secretion and interbacterial competition assays.

As suggested, we now show that a mutation in the invariant glycine of the MIX GxxY motif (G247A), which abolishes T6SS1-mediated secretion of the effector, also impairs the VPA1263-mediated intoxication of a VPA1263-sensitive prey strain during competition (Fig. 3B).

Minor comments:

1. Line 33-34, I would suggest to rephrase the statement "for the first time, that a domain used as a marker for secreted toxins is actually required for toxin secretion". In fact, I would have expected this requirement to be discovered when MIX domain was identified. Emphasis on this point seems unnecessary.

The statement was rephrased and now reads: "We also showed that a domain used as a marker for secreted toxins is required for secretion of the toxin via a type VI secretion system."

2. Data in Figure 1C showing the toxicity of various VPA1263 in E. coli is quite clear. I am curious about what is the degree of the toxicity? Although it is not essential, it would be more informative to carry out growth curve analysis to compare the degree of toxicity and how well immunity protein rescue the toxicity.

Data obtained from growth curves did not result in additional insights beyond the provided toxicity assays, and we therefore did not include it in the manuscript.

3. It is interesting that HND nuclease domain itself is not stable. I am also curious whether H837A mutation in HNH domain can stabilize this protein or this domain requires pyocin S domain for stability. The authors may test this idea or discuss this possibility.

We prefer not to draw conclusions from the apparent low levels of HNH domain expression at this time. An H837A mutation did not result in increased abundance of the HNH domain; it may be simply a matter of the choice of the start site for expression of the HNH domain alone.

Reviewer #3 (Comments for the Author):

The manuscript "A DNase T6SS effector requires its MIX domain for secretion" approach interesting questions regarding bacterial interactions mediated by T6SS. In this manuscript, the authors characterized a novel DNase effector, provided experimental evidence supporting the role of MIX domains for effector secretion, and found an association between antibacterial and anti-phage systems with a mobile element. Overall, the study is solid and supports its conclusions.

We thank the reviewer for the helpful suggestions.

My major concern is:

1: The study has three major parts (effector characterization, MIX domain, and mobile element). These sections are not well woven together. In addition, the MIX domain and mobile element sections are explored superficially instead of developing a more interesting in-depth study.

As noted above, this last section on the genomic neighborhood of vpa1263 was removed from the manuscript.

My minor concerns are:

1: Supplementary Fig. S3C data is important to support the microscopy data, and it should be in the main Figure 2

These results are similar to data that has already been published when the effector was first discovered (Salomon et al., PNAS, 2014). Therefore, since this is simply a control experiment, we think that it is appropriate as a supplementary Figure.

2: Quantification in figure 2C must have statistical tests. Also, three separate columns would be more clear than stacked columns.

To clarify the result, we have revised the relevant panel (Fig. 2C); we now only show the percentage of cells with aberrant DNA morphology, and we provide a statistical test.

3: A larger field of view of figure 2C with more cells should be added as supplemental material to show how consistent the phenotype is.

The requested wide-field view of the microscopy results is now provided as Supplementary Fig. S4. The quantitative data in Fig. 2C, summarizing the results of three biological replicates, further show that the phenotype is consistent.

4: Line 144 - Why an hns mutant was used? It should be on text.

We revised the text to clarify this point. It now reads: "...against attacker strains with a constitutively active T6SS1 deleted for the T6SS1 repressor hns to maximize T6SS1 activity (43), with or without a deletion in vpa1263 (Δ hns/ Δ vpa1263 and Δ hns, respectively)."

5: Line 147 - Is the retained the ability to intoxicate Vibrio natriegens prey in competition because other effectors can still be delivered? If so, this information should be in the text.

The revised text now reads: "...the Δ hns/ Δ vpa1263 strain retained the ability to intoxicate Vibrio natriegens prey in competition, indicating that it was able to deliver other T6SS1 effectors..."

6: Line 201 - Provide a more quantitative/explanatory definition than "nearly identical"

As noted above, this section was removed from the manuscript.

7: Line 206 - The word "edge", should be substituted for something more informative and standard for genomics, such as downstream, upstream, or flanking.

As noted above, this section was removed from the manuscript.

8: 315 - Even if it is a standard method, provide more information than citing the previous paper. This will help reproducibility.

Additional information on the deletion method is now provided.

9: Are vpa1268-vpa1270 enough to mediate the mobilization of genes? This information would support the hypothesis of mobilization of these islands across Vibrio.

As noted above, this section was removed from the manuscript.

10: Are the vpa1268-vpa1270 mobile elements conserved in other bacteria beyond Vibrio? This analysis would make the manuscript more interesting for a broader audience.

As noted above, this section was removed from the manuscript.

August 26, 2022

Dr. Dor Salomon
Tel Aviv University
Clinical Microbiology and Immunology
Tel Aviv
Israel

Re: Spectrum02465-22R1 (A DNase T6SS effector requires its MIX domain for secretion)

Dear Dr. Dor Salomon:

Thank you for submitting your revised version with the additional modifications required by the reviewers. I am pleased to inform that your manuscript has been accepted, and I am forwarding it to the ASM Journals Department for publication. You will be notified when your proofs are ready to be viewed.

Sincerely,

Ethel Bayer-Santos
Editor, Microbiology Spectrum
